# A simple, efficient and scalable contrastive masked autoencoder for learning visual representations

## Abstract

Hybrid self-supervised learning methods that combine masked image modelling and contrastive learning have demonstrated state-of-the-art performance across many vision tasks. In this work we identify a property overlooked by previous hybrid methods: they can achieve considerable efficiency improvements compared to contrastive learning, whilst still outperforming the constituent contrastive and masked image modelling training components. To demonstrate this, we introduce CAN a minimal and conceptually clean synthesis of (C) contrastive learning, (A) masked autoencoders, and (N) the noise prediction approach used in diffusion models. CAN is designed to be efficient, masking 50% of patches in *both* views, meaning that the overall FLOPs load of SimCLR is 70% higher than CAN for ViT-L backbones. Our combined approach outperforms its MAE and SimCLR constituent parts on an extensive set of downstream transfer learning and robustness tasks under both linear probe and finetune protocols, and pre-training on large scale datasets such as JFT-300M and ImageNet-21K. Code is provided in the supplementary material, and will be publicly released.

## 1 Introduction

Contrastive learning (Chen et al., 2020b) and masked image models such as MAE (He et al., 2022) employ very different learning mechanisms. The former learns to extract features that are invariant to certain semantics-preserving variations in data, while latter reconstructs missing parts of an input, thereby learning spatial statistical correlations in data. Because of this, *hybrid* methods have recently been proposed that combine aspects of both with the goal of building a reinforced and improved training mechanism (Huang et al., 2022; Tao et al., 2022). However, existing hybrid methods tend to suffer from two weaknesses compared to MAE: 1) training costs scale more poorly as model size increases, and 2) the re-introduction of complexity-increasing tricks such as multi-cropping and use of momentum updated target networks that are commonplace in contrastive learning. This increase in complexity is especially harmful to fast iteration of new models and methods given the increased adoption of web-scale training datasets (Yu et al., 2022; Radford et al., 2021; Jia et al., 2021) and the extreme accompanying costs.

In this work we introduce CAN—a hybrid contrastive masked autoencoder designed with simplicity and efficiency as priorities. In the process our aim is to demonstrate that hybrid methods are not only a promising path to improved state-of-the-art performance (as prior work has shown) but can improve feature learning without higher computation costs or more complex training recipes. As well as a minimal fusion of contrastive learning and masked autoencoders, CAN additionally uses the denoising loss that has driven advances in diffusion models (Ho et al., 2020; Song et al., 2021). This loss predicts the *noise* added to an input image, introducing negligible overheads. Denoising offers a promising third complementary learning mechanism to contrastive learning and masked autoencoding by forcing the model to learn high-frequency information, whereas autoencoder reconstructions focus on low-frequency information (Hou et al., 2017).

We show that CAN performs favourably according to key metrics: 1) performance-efficiency trade-off compared to contrastive learning and MAE, and 2) scalability to pre-training on large datasets. Indeed, CAN enjoys stronger performance than its constituent parts on their own, whilst using considerably fewer FLOPs

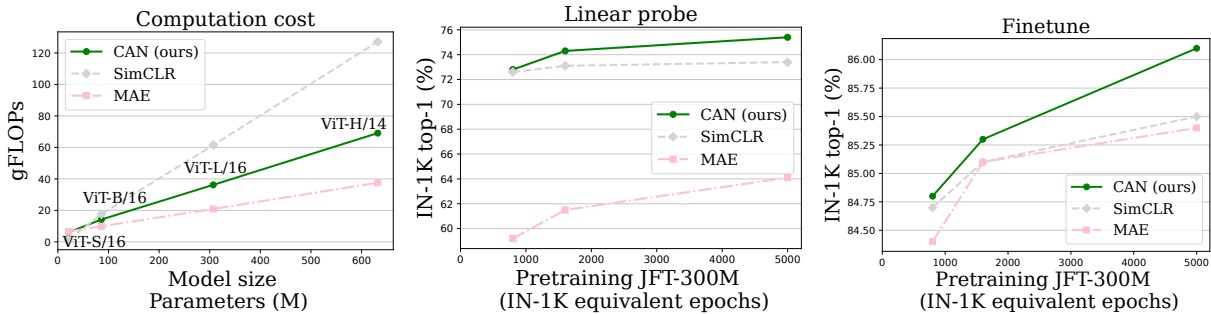

Figure 1: CAN enjoys a favourable performance-efficiency trade-off. **Left:** CAN scales more effiently than SimCLR since it uses masked inputs. **Middle and right:** CAN outperforms SimCLR and MAE on ImageNet linear probe and finetune evaluations for ViT-L models when pre-training on uncurated data such as JFT-300M.

than contrastive learning. This advantage conttinue to hold when pre-training on large datasets such as JFT-300M and ImageNet-21K, which consist of 300M and 14M images, respectively. For instance, evaluating JFT-trained ViT-L models using the top-1 accuracy of an ImageNet-trained linear probe, MAE achieves 64.1% and SimCLR achieves 73.4%, while CAN achieves 75.4%. In short, the advantages of CAN are:

1. **Simplicity.** CAN is a minimal synthesis of three powerful self-supervised learning methods: contrastive learning, masked autoencoders, and denoising.

2. **Efficiency.** CAN enjoys a favourable efficiency-performance trade-off (Figure 1), e.g., SimCLR uses 70% more FLOPs than CAN with ViT-L backbones.

3. **Scalability.** CAN scales well to training on large image datasets, such as JFT-300M and ImageNet-21K.

CAN is more efficient than SimCLR since it masks 50% of patches in each view. This also translates to faster run-times, with our largest training (ViT-L 5000 epochs) taking 2 weeks for SimCLR, and 1 week for CAN on our hardware. Our aim is to scale and solve SSL in a practical setting, specifically pre-training on large-scale datasets like JFT300M and ImageNet21k. We demonstrate that while pre-training on these large-scale datasets, we often outperform MAE, SimCLR baselines by a significant margin across 15 downstream datasets encompassing linear evaluation, fine-tuning, few-shot learning, and robustness settings.

## 2 Related Work

**Masked image models with Vision Transformers.** The advent of the Vision Transformer (ViT) (Dosovitskiy et al., 2021b) provoked a focused effort to develop strong self-supervised learning frameworks for ViT backbones. Works such as DINO (Caron et al., 2021) and MoCo-v3 (Chen et al., 2021b) demonstrated that techniques developed with ConvNet backbones in mind could also perform competitively using ViTs after proper tuning to suit the new architecture. ViT-specific methods have emerged since then, particularly masked image modelling (Bao et al., 2022; Chen et al., 2022; Xie et al., 2022), which use a mask-and-reconstruct training mechanism, taking inspiration from pre-training methods used in NLP (Devlin et al., 2018). This classical idea (Ballard, 1987) is enjoying a rejuvenation thanks to favourable efficiency when combined with the vision transformer architecture (Dosovitskiy et al., 2021b). Most notably MAE (He et al., 2022) showed that classical masked autoencoding approaches could be used to pre-train ViTs *without* passing masked tokens through the encoder. This provides a significant efficiency boost; our method similarly takes advantage of this.

**Contrastive learning in computer vision.** Self-supervision has received significant attention in computer vision as it offers a way to extract general purpose features without supervision. In particular, contrastive

learning (van den Oord et al., 2018; Hénaff et al., 2020; Chen et al., 2020b; He et al., 2020; Tian et al., 2020; Chuang et al., 2020; Hénaff et al., 2021) has achieved state of the art performance by enforcing invariance to augmentations, whilst using negative samples (Robinson et al., 2021a; Ge et al., 2021) to avoid trivial solutions by spreading the embedding out uniformly on the sphere (Wang & Isola, 2020). The contrastive pre-training task is conceptually very different from masked image models such as MAE, which learn spatial statistical dependencies. Another distinction is that autoencoders encourage information preservation in latent representations, whilst contrastive learning could suppress features (Chen et al., 2021a; Robinson et al., 2021b). This leads us to hypothesize that the two approaches learn different, complementary data features. This motivates us to combine contrastive learning and masked image modelling so as to develop a reinforced pre-training task that enjoys the merits of each.

**Denoising diffusion models.** Denoising autoencoders (DAE) (Vincent et al., 2010) learn to reconstruct clean data given a noisy input. By learning to map low-density data regions to high-density regions, DAE learns the shape of the data manifold. This connection was made precise by Vincent (2011), who showed that DAEs learn the score-function $s(\mathbf{x}) = \nabla_{\mathbf{x}} \log p(\mathbf{x})$. This key observation underpins the significant recent advances in generative diffusion models, which use an estimate of the score-function to generate samples (Ho et al., 2020; Song et al., 2021). The recent success of DAEs in generative modelling has not yet translated to representation learning, with some exceptions (Asiedu et al., 2022; Zaidi et al., 2022). In this work we exploit a denoising autoencoder to eliminate the MAE inefficiency of reconstructing unmasked patches but never using them.

**Siamese masked image modelling.** Several recent works propose approaches that combine ideas from masked image modelling and Siamese self-supervised learning. For instance, Huang et al. (2022) propose a combination of contrastive and masked reconstruction objectives using one masked view, and one full (unmasked) view. Other recent works (Tao et al., 2022; Chen et al., 2022; Assran et al., 2022) use similar asymmetric designs. The key distinction between CAN and these works is that we strike a different balance, focusing on developing a *simple*, and *efficient* method. For instance we use *two masked views* and no momentum encoder. We hope the simplicity and efficiency of CAN, and our experiments showing it's scalability, will make it easy to adapt and modify in future work.

## 3 A simple contrastive masked autoencoder framework

Our approach is a minimal synthesis of contrastive learning, the masked autoencoder (MAE) (He et al., 2022), and the denoising loss used in the training of diffusion models. We focus on simplicity and scalability, aiming to design a hybrid with as few complex or costly components as possible. We also aim to minimize *wasted* computation: in particular, the MAE decoder requires reconstructions of all patches, but only those of masked patches are used in the loss, a fact that CAN exploits. Below, first we detail the basic pipeline of generating views and passing masked inputs through the encoder and decoder, then explain the three objectives we use: contrastive, reconstruction, and denoising. The penultimate section describes the combined objective, and the final section discusses scalability.

### 3.1 Overview of method

Given a batch of $n$ images $\{\boldsymbol{x}\}_{i=1}^n$, we generate two views $\boldsymbol{x}_i^1, \boldsymbol{x}_i^2 \in \mathbb{R}^{h \times w \times 3}$ of each image without supervision using the same data augmentations as Chen et al. (2020b). Each image is then split into $T = (h/p) \times (w/p)$ non-overlapping patches of size $p \times p$: $\boldsymbol{x}_{i,\text{patch}}^1, \boldsymbol{x}_{i,\text{patch}}^2 \in \mathbb{R}^{T \times p \times p \times 3}$ in preparation for input to the ViT encoder. We always assume that $p$ divides $h$ and $w$. Two masks $\boldsymbol{M}_i^1, \boldsymbol{M}_i^2 \in \{0,1\}^T$ are independently generated, with a 1 in coordinate $t \in \{1, \ldots T\}$ indicating that the $t$-th patch is masked. Each patch is masked independently with probability $r$, conditioned on always having exactly $T' = r \cdot T$ patches masked, which we assume is an integer. In all CAN experiments our default masking rate is $r = 50\%$ unless explicitly stated otherwise (note that for all MAE results we follow the exact settings as in (He et al., 2022) using the default $r = 75\%$). Following He et al. (2022), only the $T - T'$ *unmasked* patches are passed to the ViT encoder, which processes the two views in parallel. Masking a large fraction of patches from both views makes our method much more efficient (see Table 1) than contrastive methods that use two full views and recent works that use one full view and one masked view (Assran et al., 2022; Huang et al.,

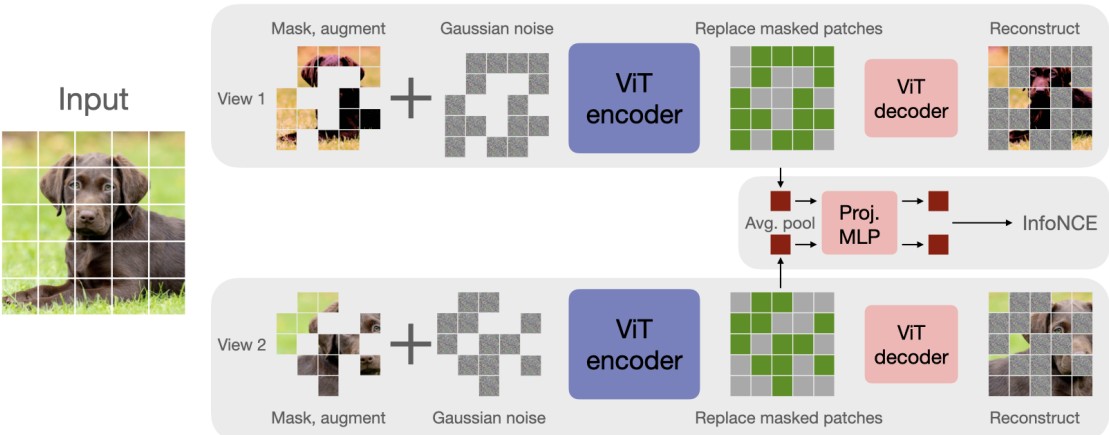

Figure 2: **The CAN framework:** Two views of an image are generated, 50% of patches randomly masked in each, and noise is added to patches. An encoder is trained to solve three tasks: 1) **Reconstruction:** encoded patches are passed to a decoder that reconstructs missing patches, 2) **Denoise:** reconstructs the noise added to unmasked patches, and 3) **Contrast:** pooled patches are passed to a contrastive loss, using in-batch samples as negatives (Chen et al., 2020b).

2022). Finally, we collect the embeddings of unmasked tokens $\boldsymbol{bz}_i^1, \boldsymbol{bz}_i^2 \in \mathbb{R}^{(T-T')\times d}$ and reshape into $T \times d$ tensors by adding a learned [M] embedding to positions corresponding to masked tokens. The result is passed through a comparatively lightweight ViT decoder to produce outputs $\hat{\boldsymbol{bx}}_i^1, \hat{\boldsymbol{bx}}_i^2$ in image space $\mathbb{R}^{h\times w\times 3}$.

## 3.2 Contrastive learning objective

The embeddings $\boldsymbol{bz}_i^1, \boldsymbol{bz}_i^2 \in \mathbb{R}^{(T-T')\times d}$ returned by the encoder are pooled via a simple mean along the first dimension to form $d$-dimensional embeddings, which are passed through a lightweight MLP projection head that maps into a lower dimension space $\mathbb{R}^r$, $r < d$, and normalized to unit length to produce embeddings $\boldsymbol{bu}_i^1, \boldsymbol{bu}_i^2 \in \mathbb{R}^r$ for $i = 1, \ldots n$. For the $i$th batch item we collect the other $2n - 2$ samples in-batch $\mathcal{N}_i = \{\boldsymbol{bu}_j^1, \boldsymbol{bu}_j^2\}_{j\neq i}$ to use as negatives, and compute the $\mathcal{L}_{\text{InfoNCE}}$ loss:

$$\frac{1}{2n}\sum_{v=1,2}\sum_{i=1}^{n} -\log \frac{e^{\boldsymbol{bu}_i^{1\top}\boldsymbol{bu}_i^2/\tau}}{e^{\boldsymbol{bu}_i^{1\top}\boldsymbol{bu}_i^2/\tau} + \sum_{\boldsymbol{bu}^-\in\mathcal{N}_i} e^{\boldsymbol{bu}_i^{v\top}\boldsymbol{bu}^-/\tau}}$$

where $\tau > 0$ is a temperature parameter, defaulting to 0.1. Our choice of InfoNCE objective is justified by recent work (Koppula et al., 2022) that found that a simple InfoNCE objective as in SimCLR scales to large dataset better than methods such as BYOL (Grill et al., 2020) or DINO (Caron et al., 2020).

## 3.3 Patch reconstruction objective

The outputs $\hat{\boldsymbol{bx}}_i^1, \hat{\boldsymbol{bx}}_i^2$, $i = 1, \ldots, n$ of the ViT decoder are trained to reconstruct the missing patches of each image. As in He et al. (2022), we find it best to only compute the reconstruction loss on masked patches:

$$\mathcal{L}_{\text{rec}} = \frac{1}{2n}\sum_{v=1,2}\sum_{i=1}^{n} \|\boldsymbol{bM}_i^v \circ (\boldsymbol{bx}_i^v - \hat{\boldsymbol{bx}}_i^v)\|_2^2$$

where $\circ$ multiplies all pixels in the $t$th patch of the residual image $\boldsymbol{bx}_i^v - \hat{\boldsymbol{bx}}_i^v$ by $(\boldsymbol{bM}_i^v)_t \in \{0, 1\}$.

Whilst computing the loss only on masked patches gives better performance, it indicates wasted computation since the decoder also produces reconstructions for unmasked patches. To avoid waste we propose an alternative objective specifically for unmasked patches, which we discuss next.

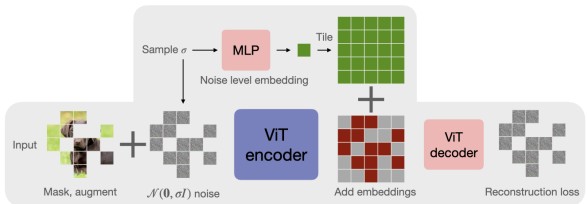

Figure 3: **Denoising:** Both the encoded patches and the noise level $\sigma$ are passed to the decoder by passing $\sigma$ through an MLP, and adding the result to each embedded token.

### 3.4 Denoising objective

Inspired by the significant advances in diffusion modelling using *denoising* training objectives (Ho et al., 2020; Kingma et al., 2021) and their equivalent score-based counterparts (Song et al., 2021; Vincent, 2011) we revisit the suitability of denoising for self-supervised learning. We add independent isotropic Gaussian noise to each image $\boldsymbol{b}x_i^v \leftarrow \boldsymbol{b}x_i^v + \sigma_i^v \boldsymbol{b}e_i^v$ with $\boldsymbol{b}e_i^v \sim \mathcal{N}(\boldsymbol{b}0, I)$ and $\sigma_i^v$ uniformly sampled from an interval $[0, \sigma_{\max}]$. This noisy input is masked and passed to the encoder as described in Section 3.1. When passing encoded patches to the decoder we make a small addition to the method in Section 3.1 to provide the decoder with information on the noise level $\sigma_i^v$ to help it separate noise from the ground truth image. This is motivated by denoising diffusion methods, which pass both the noisy image and the noise level as inputs to the denoising model (Ho et al., 2020). We approach this by using $\sigma_i^v$ as a positional encoding in the decoder, similarly to Vaswani et al. (2017). First we produce a sinusoidal embedding of $\sigma_i^v \in \mathbb{R}^d$, which is passed through a lightweight 2 layer MLP with ReLU activations of constant width $d$ to produce a (learnable) embedding $\boldsymbol{b}p_i^v \in \mathbb{R}^d$, whose dimension matches the latent dimension of $\boldsymbol{b}z_i^v \in \mathbb{R}^{T \times d}$. We add the result to each embedded token (including missing tokens [M]) to provide noise-level information: $(\boldsymbol{b}z_i^v)_t \leftarrow (\boldsymbol{b}z_i^v)_t + \boldsymbol{b}p_i^v$ for $t = 1 \ldots, T$, and pass the result to the decoder producing $\hat{\boldsymbol{b}x}_i^v$. We define our denoising loss function, which is computed only on unmasked pixels:

$$\mathcal{L}_{\text{denoise}} = \frac{1}{2n} \sum_{v=1,2} \sum_{i=1}^n \|(1 - \boldsymbol{b}M_i^v) \circ (\sigma_i^v \boldsymbol{b}e_i^v - \hat{\boldsymbol{b}x}_i^v)\|_2^2$$

where, $\circ$ multiplies pixels by the patch-level masking as in Section 3.3. Note that the reconstruction loss $\mathcal{L}_{\text{rec}}$ still uses the *clean* input $\boldsymbol{b}x$ as its target, with no noise added. The denoising loss is extremely lightweight, introducing only a very small overhead due to the MLP. We emphasize that the reconstruction of noise patches comes at zero additional cost since the decoder produces reconstructions of all patches, both masked and unmasked, but only reconstructions of masked patches are used in $\mathcal{L}_{\text{rec}}$. Finally, it has been observed in the diffusion modelling literature that although it is equivalent to train a denoising model to estimate the noise $\boldsymbol{b}e$, or to estimate the clean input $\boldsymbol{b}x$ (Vincent, 2011), there is an empirical gap, with noise target faring better. While we do not pursue it further, our testing corroborates this.

### 3.5 The combined objective function

The overall CAN objective trains the encoder and decoder to optimize three losses combined:

$$\mathcal{L}_{\text{CAN}} = \lambda_{\text{InfoNCE}} \mathcal{L}_{\text{InfoNCE}} + \lambda_{\text{rec}} \mathcal{L}_{\text{rec}} + \lambda_{\text{denoise}} \mathcal{L}_{\text{denoise}}$$

where $0 \leq \lambda_{\text{InfoNCE}}, \lambda_{\text{rec}}, \lambda_{\text{denoise}}$, and $\lambda_{\text{InfoNCE}} + \lambda_{\text{rec}} + \lambda_{\text{denoise}} = 1$ weight the objectives. In practice we parameterize the weights by eliminating one variable using the equality constraint, taking: $\lambda_{\text{rec}} = (1 - \lambda_{\text{InfoNCE}}) \cdot \lambda$ and $\lambda_{\text{denoise}} = (1 - \lambda_{\text{InfoNCE}}) \cdot (1 - \lambda)$ where $0 \leq \lambda \leq 1$. This parameterization makes it easy to control the weighting between the two reconstruction losses $\mathcal{L}_{\text{rec}}, \mathcal{L}_{\text{denoise}}$ on the one hand, and the contrastive loss $\mathcal{L}_{\text{InfoNCE}}$ on the other. We find that performance is robust to the choice of $\lambda$, and many choices of $\lambda_{\text{InfoNCE}}$ also work well (see Section 5).

|  | Architecture | Epochs | IN-1K top-1 |
|---|---|---|---|
| MoCLR (Tian et al., 2021) | R50 | 5000 | 67.6 |
| BYOL (Grill et al., 2020) | R50 | 5000 | 67.9 |
| DnC (Tian et al., 2021) | R50 | 1000 | 67.9 |
| DnC (Tian et al., 2021) | R50 | 4500 | 70.7 |
| MoCLR (Tian et al., 2021) | R200×2 | 5000 | 74.2 |
| DnC (Tian et al., 2021) | R200×2 | 3000 | **77.3** |
| MAE† (He et al., 2022) | ViT-L | 1600 | 50.5 |
| MAE† (He et al., 2022) | ViT-L | 5000 | 64.1 |
| SimCLR† (Chen et al., 2020b) | ViT-B | 800 | 65.8 |
| SimCLR† (Chen et al., 2020b) | ViT-L | 800 | 72.6 |
| SimCLR† (Chen et al., 2020b) | ViT-L | 1600 | 73.1 |
| SimCLR† (Chen et al., 2020b) | ViT-L | 5000 | 73.4 |
| **CAN (ours)** | ViT-B | 800 | 67.1 |
| **CAN (ours)** | ViT-L | 800 | 72.8 |
| **CAN (ours)** | ViT-L | 1600 | 74.3 |
| **CAN (ours)** | ViT-L | 3000 | 75.3 |
| **CAN (ours)** | ViT-L | 5000 | 75.4 |

Table 1: **JFT-300M pre-training:** Comparison to the state of the art on ImageNet linear probe. CAN outperforms all methods except DnC, which uses a complicated multi-stage training process. Computation is measured as ImageNet-equivalent epochs. †Our implementation of (Chen et al., 2020b) and (He et al., 2022).

### 3.6 Discussion on Efficiency

The efficiency of CAN arises from masking 50% of both views. We also omit certain design choices in the interests of efficiency: we do not use a momentum encoder or multiple views (multi-crop). Each of these components tends to add significant (2× or more) expense to training. Even without these components CAN achieves strong performance, outperforming its key constituent parts SimCLR and MAE.

## 4 Results

### 4.1 Pre-training on uncurated data: JFT-300M

A key promise of self-supervised learning is to allow models to be trained on extremely large scale image datasets collected from the Web. Not only is such data likely to be *unannotated*, but also *uncurated*: images containing many objects, variable lighting, artifacts (e.g., watermarks) and so on. The large variation in images found online presents a major challenge to self-supervised learning, and it is not guaranteed that methods that work well on curated (and comparatively smaller) datasets such as ImageNet will work equally well on less curated data. To study how CAN scales to large datasets we use JFT-300M (Sun et al., 2017), a dataset of around 300 million images.

**Setup.** Training time is measured in ImageNet-equivalent epochs: 1 epoch equals 1281167/[batch size] steps, the number of steps in one IN-1K epoch. Models are evaluated using linear probe and finetuning on IN-1K. All hyperparameers were tuned on IN-1K, besides learning rate and weight decay which we cut by a factor of 4 and 2 respectively to stabilize training on JFT-300M. See Appendix C and Section 5 for details.

**Results.** Figure 1 compares CAN to SimCLR and MAE baselines using ViT-L models. CAN achieves a much better trade-off between efficiency (measured in FLOPs) and performance using ViT-L models for all three methods: SimCLR uses 70% more FLOPs than CAN, which consistently outperforms both SimCLR and MAE: for training ViT-L models for 5000 epochs, CAN achieves an IN-1K linear probe performance of 75.4%, compared to 73.4% for SimCLR and 64.1% for MAE. The relatively poorer linear probe performance of MAE on JFT-300M highlights the non-triviality of scaling from IN-1K to larger datasets and suggests that

| Method | Pre-training epochs | Encoder | No Additional params. | Masked image | Finetune | Linear probe |
|---|---|---|---|---|---|---|
| *from scratch* | 100 | ViT-B | ✓ | ✗ | 79.1 | — |
| MoCo-v3 (Chen et al., 2021b) | 300 | ViT-B | ✗ | ✗ | 83.0 | 76.7 |
| DINO (Caron et al., 2021) | 1600 | ViT-B | ✗ | ✗ | 82.8 | 78.2 |
| CIM (Fang et al., 2022) | 300 | ViT-B | ✗ | ✗ | 83.1 | — |
| CAE (Chen et al., 2022) | 800 | ViT-B | ✗ | ✗ | 83.8 | 68.6 |
| CAE (Chen et al., 2022) | 1600 | ViT-B | ✗ | ✗ | 83.9 | 70.4 |
| BEiT (Bao et al., 2022) | 800 | ViT-B | ✗ | ✗ | 83.2 | 37.6* |
| SimMIM (Xie et al., 2022) | 800 | ViT-B | ✓ | ✗ | 83.8 | 56.7 |
| MAE (He et al., 2022) | 800 | ViT-B | ✓ | ✓ | 83.1 | — |
| MAE (He et al., 2022) | 1600 | ViT-B | ✓ | ✓ | 83.6 | 68.0 |
| **CAN (ours)** | 800 | ViT-B | ✓ | ✓ | 83.4 | 74.0 |
| **CAN (ours)** | 1600 | ViT-B | ✓ | ✓ | 83.6 | 74.8 |
| SimCLR† (Chen et al., 2020b) | 800 | ViT-L | ✓ | ✗ | 83.4 | 73.9 |
| MAE (He et al., 2022) | 800 | ViT-L | ✓ | ✓ | 84.9 | 73.5 |
| MAE† (He et al., 2022) | 800 | ViT-L | ✓ | ✓ | 83.7 | 71.4 |
| **CAN (ours)** | 800 | ViT-L | ✓ | ✓ | 84.7 | 76.2 |

Table 2: **Finetune and linear probe results with pre-training on ImageNet-1K.** Note that CAN does not use multi-crop augmentation or momentum encoder. †Our implementation of (Chen et al., 2020b) and (He et al., 2022). *Quoted from Chen et al. (2022).

while MAE is scalable for *model size*, scalability to larger *datasets* requires further study. Figure 1 (right) gives finetuning results. CAN performs favourably: for a 5000 epoch pre-training schedule, CAN achieves an IN-1K linear probe performance of 86.1%, compared to 85.5% for SimCLR and 85.4% for MAE. CAN also enjoys better scaling with training schedule length than either MAE or SimCLR, with the difference in performance becoming *larger* for longer schedules. We hypothesize that this is not coincidental, and that strong pre-training tasks like CAN play an important role in scalability.

We also compare CAN to the current state of the art on JFT-300M pre-training in Table 1. Our best performance, 75.4% with ViT-L outperforms all methods besides DnC, with 77.3% (Tian et al., 2021) with R200×2. However we note that CAN is *considerably* simpler than DnC, which involves training 10 separate "expert" models (each as large as the final model), and then using MoCLR (an improvement of SimCLR that adds a momentum encoder and more), using distillation to produce a single final model. Our calculations suggest that training a ViT-L with CAN is about 3× faster than training the considerably smaller ResNet50 with DnC in terms of wall clock time (see Appendix B for explanation). CAN on ViT-L outperforms MoCLR with R200×2 backbone (similar parameter counts), where we note that MoCLR performs as well or better than BYOL and MoCo-v3 on IN-1K (Tian et al., 2021).

### 4.2 Pre-training on ImageNet-21K

We also consider the performance of CAN on pre-training on ImageNet-21K (IN-21K), a publicly available dataset of 14.2 million images Deng et al. (2009). We use the same hyperparameter settings as JFT-300M. We run a full set of evaluations on linear probe (Table 6), robustness (Figure 15), and few-shot learning (Figure 16) (see Sections 4.4 and 4.5 for details on few-shot and robustness evaluations). Results are reported in Appendix A.1. CAN also performs well with IN-21K pre-training, with CAN finetuned on IN-1K showing better robustness than MAE and SimCLR in 8 out of 8 cases, and CAN achieving best 25-shot performance on 6 out of 9 datasets.

### 4.3 Pre-training on ImageNet-1K

Next we evaluate our method using ImageNet (IN-1K) pre-training to verify that it is also competitive in this setting. Results in Table 2 record the top-1 accuracy on IN-1K classification of finetuned models and linear probes. Finetuning CAN achieves 83.6% with ViT-B, outperforming other contrastive approaches such as MoCo-v3 (83.0%), and is competitive with other state-of-the-art approaches such as CAE (83.9%). The linear probe performance of CAN is 74.8% using ViT-B, beating all masked image modelling methods, the best of which is CAE with 70.4% (Chen et al., 2022). CAN is only outperformed by MoCo-v3 and DINO, which use momentum encoders and two full image views, and in the case of DINO 10 multi-crop views. Note

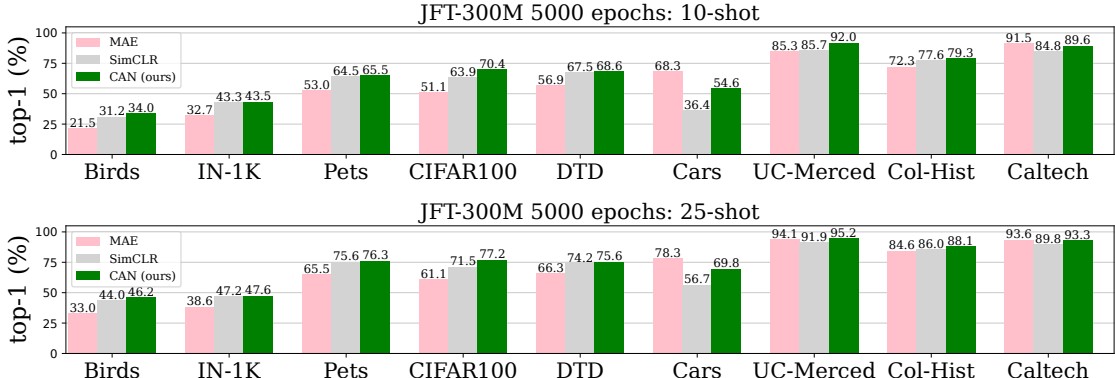

Figure 4: **Few-shot:** ViT-L models pre-trained on JFT-300M for 5000 epochs are evaluated on 9 datasets in few-shot setting (10-shot and 25-shot). CAN outperforms MAE and SimCLR.

that the *masked image* column indicates whether a method uses one or more full image views as input to the model, and the *no additional parameters* column indicates whether a method relies on other parameters besides the main encoder, e.g., from a pre-trained tokenizer, or a momentum updated target encoder. We also report results for our MAE implementation, which approximately matches the numbers reported in He et al. (2022), validating our MAE results on JFT-300M.

### 4.4 Few-shot learning

We use linear probes to evaluate suitability of CAN for few-shot learning, following the protocol of Dosovitskiy et al. (2021a). We use the models pre-trained on JFT-300M for 5000 epochs whose ImageNet performance is recorded in Figure 1. Results in Figure 4 for few-shot transfer learning on 9 other datasets show that the superior performance on IN-1K translates to strong performance on other tasks. We also note that our 25-shot ViT-L models beat *full-shot* both DnC and BYOL ResNet50 models (also trained for 5000 epochs on JFT-300M) on 6 out of 8 datasets (Tian et al., 2021). See Appendix A for many additional results, including pre-training on IN-21K.

### 4.5 Robustness to distribution shift

Finally, we consider the robustness of CAN to distribution shifts. We use ViT-L backbones trained for 5000 epochs on JFT-300M, which have been finetuned on IN-1K. Model performance is evaluated on a number of different validation sets with the same 1000 classes as IN-1K Mao et al. (2022). Figure 5 reports results on the following 7 validation sets, which cover a large variety of distribution shifts: original IN-1K (Deng et al., 2009), IN-v2 (Recht et al., 2019), IN-ReaL (Beyer et al., 2020), IN-Adversarial (Hendrycks et al., 2021b), IN-Rendition (Hendrycks et al., 2021a), ObjectNet (Barbu et al., 2019). CAN performs favourably under both JFT-300M, IN-21K and IN-1K pre-training, beating SimCLR and MAE baselines in nearly all cases. See Appendix A for additional results.

## 5 Hyperparameter analysis

We study the different components of CAN to better understand the effect of the different mechanisms, and to determine optimal parameter configurations. All ablations use ViT-B models trained for 100 epochs on IN-1K and evaluated with a linear probe on IN-1K unless explicitly said otherwise. We use the best loss weights

| Method | Contrastive loss ↓ | Reconstruction loss ↓ |
|---|---|---|
| SimCLR | 9.157 | — |
| MAE | — | 0.1658 |
| **CAN (ours)** | **9.143** | **0.1633** |

Table 3: **Loss complementarity.** CAN training achieves *lower* training loss for both contrastive and reconstruction than individual training. All methods use 50% masking for fair comparison.

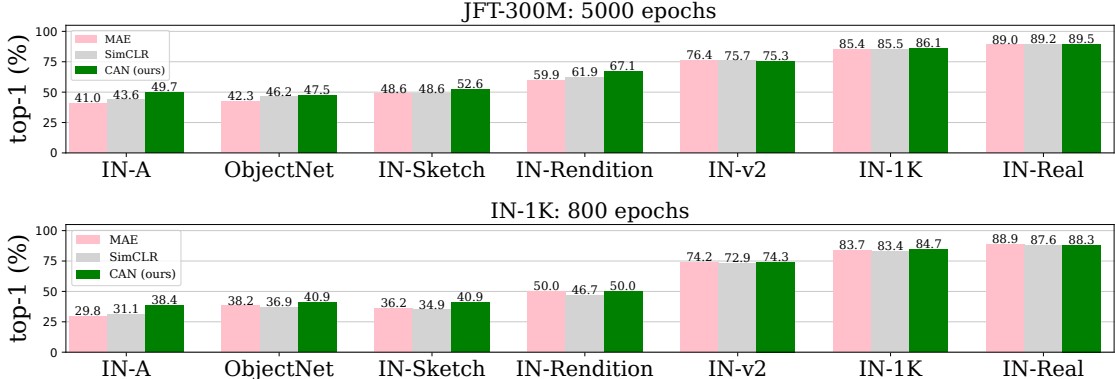

Figure 5: **Robustness:** Evaluating performance under distribution shifts with respect to models finetuned on IN-1K. Validation performance of ViT-L models is reported on 7 different datasets.

and noise level in these experiments for experiments in Section 4.

**Complementarity of contrastive and reconstruction losses.** A key hypothesis motivating our work is that contrastive learning and masked autoencoder reconstruction may not only be compatible training objectives, but are *complementary* ones. Table 3 compares the final training value of the contrastive $\mathcal{L}_{\text{InfoNCE}}$ and reconstruction $\mathcal{L}_{\text{rec}}$ when jointly trained (i.e., CAN) compared to only optimizing $\mathcal{L}_{\text{InfoNCE}}$ (SimCLR) or only $\mathcal{L}_{\text{rec}}$ (MAE). The results support the hypothesis: joint training achieves a lower loss on *both* objectives compared to individual training.

| None | +noise | +noise, +loss | Full |
|------|--------|---------------|------|
| 67.9 | 68.6   | 68.4          | 68.9 |

Table 4: **Denoising objective.** "Full" denotes the entire method as described in Section 3.4

| AN | CN | CA | CAN (full) |
|------|------|------|------------|
| 42.8 | 68.5 | 67.9 | 68.9 |

Table 5: **CAN loss terms.** We remove each of the three loss terms in CAN one by one.

**Ablating CAN loss terms.** CAN is comprised of three components: (C) contrastive, (A) masked autoencoder, and (N) denoising losses. We ablate each of the three components in Table 5, setting the loss weight to zero to "remove" a component. We use ViT-B models pre-trained for 100 epochs. Removing any component leads to worse performance, with contrastive loss hurting the most.

**Denoising method.** Table 4 studies the effect of each of the components of the denoising method. We use ViT-B models trained for 100 epochs on ImageNet, and consider four settings, each adding in more parts of the method: 1) CAN with no denoising, 2) adding noise to the input only, 3) adding noise and using the denoising loss, and 4) the full method with all of the described components, including using $\sigma_i^v$ as a positional encoding in the decoder. Results show that simply adding noise as a data augmentation improves performance by 0.7%, which can be improved to 1% by adding a reconstruction loss with noise level passed as an argument. The noise level argument is necessary: the reconstruction loss without noise level argument performs worse (68.4%) than noise with no reconstruction at all (68.6%). We emphasize that the improvement from denoising comes at minimal run time and memory cost, since it uses reconstructions produced by the decoder, which in the case of MAE are simply thrown away unused. We also tried predicting the clean patches instead of noise, and found it worked poorly, corroborating similar findings in the diffusion literature.

**Masking rate.** Figure 6 reports the behavior of CAN and SimCLR under different masking rates on IN-1K and JFT-300M pre-training (for JFT-300M we use 800 epochs). The performance of SimCLR decreases as the masking rate increases, suggesting that masking is not an effective data augmentation. In contrast, performance of CAN peaks at a non-zero masking rate, but at a much lower rate than the 75% used by MAE on IN-1K. This occurs since very low masking rates are preferred by the contrastive part of CAN, but severely damage the autoencoder part as it can learn trivial solutions. The considerable efficiency improvement from

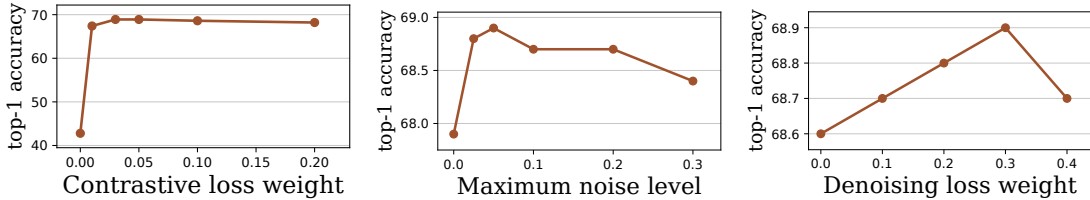

Figure 6: CAN and SimCLR with different masking rates. ViT-B models are pre-trained for 100 epochs on IN-1K (left), and 800 epochs on JFT-300M (right).

Figure 7: ViT-B models pre-trained on IN-1K for 100 epochs. **Left:** The best contrastive loss weight is small but non-negative. **Middle:** A wide range of $\sigma_{\max}$ values improve over no-noise. **Right:** Performance is not sensitive to the denoising loss weight.

masking 50% of patches more than compensates for the small drop in performance for a fixed number of epochs.

**Contrastive loss weight.** We vary the weighting $\lambda_{\text{InfoNCE}}$ used to weight the contribution of the contrastive and reconstruction losses. Recall that larger $\lambda_{\text{InfoNCE}}$ places higher weight on the contrastive loss. Results in Figure 7 show that the best weight is $\lambda_{\text{InfoNCE}} = 0.03$, which approximately balances the magnitudes of the two terms (see Table 3).

**Denoising loss weight and noise level.** We study the noise level interval $[0, \sigma_{\max}]$ from which to sample input noise, and the weight $\lambda$ balancing the denoising and reconstruction losses. Results in Fig. 7 show that the best maximum noise level is $\sigma_{\max} = 0.05$, and that similar performance is attained for different weights on the denoising loss.

## 6 Discussion

We present CAN, a simple, efficient and scalable self-supervised method for visual representation learning. CAN combines ideas from contrastive learning, masked autoencoding, and diffusion denoising into a single high-performing method. Extensive empirical results show that CAN scales with minimal changes to the large uncurated datasets, outperforming SimCLR and MAE methods on a wide range of downstream tasks and evaluations, including ImageNet linear probes, few-shot, robustness, and finetuning. Our results suggests that combining different self-supervised methods can produce better results than the constituent parts alone. Further exploration of this search space appears a promising avenue for future work.

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
