# OpenReview forum: "A simple, efficient and scalable contrastive masked autoencoder for learning visual representations"
_TMLR — Rejected by TMLR_

### Review · Reviewer_cD6G · 2023-07-12

**Summary Of Contributions:**

The authors propose a new contrastive-MAE-hybrid self-supervised representation learning method, CAN, which is short for (C) contrastive learning, (A): masked autoencoders, and (N) noise prediction. The key idea is to combine contrastive objectives with masked autoencoders by masking argument views, where a noise prediction task is further used to reinforce the high-frequency focus of MAEs. CAN is very simple. Extensive experiments on image classification with impressive results demonstrate that CAN is more training-efficient and scalable to the pretraining dataset scale.

**Audience:**

Yes

**Broader Impact Concerns:**

The broader impact is not discussed, but I do not have any critical concerns regarding this.

**Claims And Evidence:**

Yes

**Requested Changes:**

The requested changes are mainly described in the pros and cons part. Besides the revisions or clarifications requested before, there are some experiments to be conducted to make the work more solid, but I would suggest some actions as follows:

- Object detection experiments on COCO and semantic segmentation results on ADE20K.
- Possible experiments to extend CAN to other (multimodal) contrastive learning methods.

**Strengths And Weaknesses:**

### Strength
- This work focuses on developing training-efficient and scalable self-supervised representation learning with a contrastive-MAE-hybrid fashion, where visual pretraining on large-scale datasets like JFT-300M are explored. The targeted problems of performance-efficiency trade-off, scalability, and hybrid pretraining are critical and relevant. In modern AI developments, this is very important when considering large-scale uncurated data from sources like the web.
- The proposed method is simple, clear, and easy to follow. The idea of combining MAE and contrastive learning is relatively straightforward, while the idea of introducing denoising objectives for improving high-frequency focus is interesting to me.
- Extensive experiments on image classification have been conducted, and impressive results have been achieved.

### Weakness
- My first **major concern** lies in the technical novelty contribution. As mentioned in the Strength, CAN is quite simple without complicated designs. However, this does not change the fact that the major modifications to combine contrastive learning and MAE upon baseline methods like SimCLR is to add masks and the MAE objective to argument views. Masking strategy has been proven to improve training efficiency [Li et al., 2023; Yang et al., 2023], CAN seems quite straightforward for a hybrid representation learning. However, I like the idea of denoising prediction for further improvement (which makes sense but is not significant, though).
- My second **major concern** is about the experimental evaluation. The current experiments are conducted on large-scale datasets like JFT-300M, which is good (but an in-house dataset that is not available to the community). However, the main evaluation on downstream tasks only includes ImageNet-1K classification. As a foundation representation learning method, it is necessary to evaluate its representation transferring performance on tasks like object detection and semantic segmentation since IN-1K Top-1 or linear probing performance cannot reveal all aspects of the representation nature.
- As mentioned before, the method's modification is mainly masks and MAE objective. Tab. 4-5 and Fig. 6 show that denoising prediction is not the essential part of the designs since only a marginal improvement is achieved. Besides, since contrastive learning and MAE baselines are critical, the results should be included in Tab. 5 for a more comprehensive comparison. From the result of AN, I would say that the MAE performance is very low, which is confusing and should be explained.
- The main results are based on one contrastive baseline, SimCLR. I wonder about the performance of CAN to other methods like VICReg [Bardes, Ponce & LeCun, 2022].
- What about extending to multimodal contrastive learning? For example, CLIP?
- Why are the ImageNet-equivalent epochs a fair metric for the training efficiency comparison? Why not directly report the training GPU hours? It is a bit strange to me.
- Can authors provide more insights into the method like some theoretical evidence? Why is denoising prediction helping MAE since MAE itself can be viewed as denoising autoencoding (DAE)?
- Missing citations and comparisons besides the works mentioned before. This paper focuses on hybrid SSL. However, no comparisons to prior arts discussed in related work have been included in the main results. Besides, some recent work on hybrid SSL is not compared or discussed [Jing, Zhu & LeCun, 2022; Qi et al., 2023].
- Minor: All appendix references are ???. The writing should be improved. For example, for the experimental discussion, include more details and analysis which may help strengthen the insights of this work.

[Li et al., 2023] Scaling Language-Image Pre-training via Masking. In CVPR.

[Yang et al., 2023] Attentive Mask CLIP. arXiv preprint.

[Bardes, Ponce & LeCun, 2022] VICReg: Variance-Invariance-Covariance Regularization For Self-Supervised Learning. In ICLR.

[Jing, Zhu & LeCun, 2022] Masked Siamese ConvNets. arXiv preprint.

[Qi et al., 2023] Contrast with Reconstruct: Contrastive 3D Representation Learning Guided by Generative Pretraining. In ICML.

---

> ### Author Response · Authors · 2023-08-20
> **New object detection results**
>
> We are sincerely grateful for the time volunteered to review our work. We are glad you appreciate several strengths of our work, including the improved efficiency-performance trade-offs of CAN, the simplicity of the approach, and the extensive evaluations.
>
> Your review also raised a few concerns, which we are glad to have the chance to discuss further.
>
> ---
>
> > Technical novelty of CAN
>
> You are right, there already exist multiple hybrid contrastive masked-autoencoder methods. We try to be as clear about this as possible, discussing the literature in the introduction, and related works.
>
> **Contrastive methods with masking**
>
> We like these works a lot, and view our work as continued exploration of this line of thinking.
>
> Points to make: we explore more design choices such as a zero-overhead denoising loss, and a light-weight decoder for reconstruction. Anything further?
>
> **Hybrid contrastive masked-autoencoders**
>
> The observation that distinguishes CAN from prior hybrid methods is that hybrid methods not only can produce better performance, but also a better performance-efficiency trade-off. This has largely been missing from prior literature, which has mostly pursued state-of-the-art performance. The exception to this is the works discussed above on contrastive learning with masking, which does consider efficiency (as in Li et al. 2023), and as discussed above we view our work as continued work in this direction.
>
> ---
>
> > Empirical evaluation datasets and downstream tasks.
>
> We agree that more fine-grained image tasks are also important and valuable downstream tasks to evaluate our models on. For this reasons we ran evaluations of our models on object detection and semantic segmentation, which produced the following results:
>
>
> | Method | AP_box | AP_mask |
> |--------|--------|---------|
> | MAE  | 48.1   |      41.4   |
> | CAN    | 49.1   | 42.3    |
>
>
> For object detection and semantic segmentation we pre-train our method CAN and baseline MAE for 50 epochs using the same hyperparameters as in our paper. We then follow the same benchmark pipeline as the MAE paper for detection and segmentation. We are working on longer training runs.
>
> Although we wholeheartedly agree that object detection is valuable, which is why we ran these experiments, we would like to emphasize the large number of different evaluations on few-shot, robustness to distribution shifts, finetuning and linear probes that we gathered, and hope you find these interesting too.

---

### Review · Reviewer_htve · 2023-07-18

**Summary Of Contributions:**

This paper presents an approach that combines several self-supervised learning methods into one framework. Specifically, the proposed representation learning method involves combining masked auto-encoders, contrastive learning and image denoising objectives. In contrast to existing methods, the proposed method is designed with a focus on training efficiency to allow scalability to large datasets.

**Audience:**

Yes

**Claims And Evidence:**

No

**Requested Changes:**

See weakness above.

**Strengths And Weaknesses:**

## Strengths
- The motivation for the problem addressed in this work is clearly explained and easy to follow. In order to fully take advantage of self-supervised learning, scalability is a crucial component. The proposed work presents an approach that is efficient while taking advantage of the state-of-the-art ideas in this domain.

- The text of the paper is very well written and presents concise descriptions of the key ideas. While the idea of combining contrastive learning and masked image modeling is not novel, this work makes several novel proposals to improve efficiency. The proposed ideas of masking multiple views, removing momentum encoders, multi-crop and integrating denoising objectives are all novel ideas that can provide useful guidance for future work.

- The quantitative evaluation of the representation learned by CAN demonstrates clear superiority over all the component self-supervised learning objectives on multiple benchmarks. The experiments cover multiple datasets and multiple base architectures demonstrating a convincing increase in performance. The ablative studies presented also provide useful insights into the contribution of the 3 components and design choices proposed in this hybrid self-supervised learning approach.

## Weakness
I see only one major weakness of the proposed work. The entire text of the paper focuses on the improved efficiency of the proposed hybrid self-supervised learning approach. However, the paper doesn't present much experimental evidence about the claimed efficiency. Figure 1 demonstrates that the proposed approach is less efficient than MAEs and more efficient than SimCLR. But there is no other experiment investigating the efficiency of the approach.

The experiments section focuses heavily on demonstrating the accuracy gains obtained by leveraging the proposed approach. Since the main claim is around efficiency, I hoped to see similar experiments with regards to efficiency. For example, there is no ablation on the computational cost of the design choices made in the proposed approach.

The authors also clearly state and cite works to show that this is not the first hybrid self-supervised learning approach that combines MAEs and contrastive learning. The main contribution of this work is claimed to be a *more* efficient hybrid self-supervised learning approach. This claim is not backed by any experimental evidence. I think it is important to compare the efficiency of existing hybrid approaches to the proposed approach.

---

> ### Author Response · Authors · 2023-08-20
> **Discussion of efficiency measurements**
>
> Thank you for taking the time to review our work. We are sincerely grateful for your feedback, and are glad you appreciated several aspects of our work, including its motivation, quantitative evaluation, and written communication.
>
> Your main concerns focus on understanding the efficiency improvements of CAN. We agree that this is an important point, and are glad to further discuss the comparisons that can be made here. We have some key points for your consideration.
>
> ---
>
> > CAN is “made up of” SimCLR and MAE, so comparing these two methods is most directly natural.
>
> The choice to compare CAN to SimCLR and MAE is not arbitrary. CAN consists of a contrastive part, and an autoencoder part, and Figure 1 shows that CAN is considerably more efficient than its contrastive component, and better performing than both constituent parts. We expect the same basic observation to hold when swapping in other contrastive or reconstruction-based approaches. This Figure 1 communicates the main point we are trying to make, which is that hybrid methods are more efficient and better performing than their constituent parts if they are put together correctly.
>
> ---
>
> > The effect of intentionally omitted components is already measured in prior work.
>
> For multi-crop, the DINO paper reports that using 2 full (224x224) views and 10 smaller (96x96) views leads to a 60% increase in memory footprint and runtime. The increase in footprint will be very similar in our setting.
>
> We also manually measure the effect of a momentum encoder on memory usage. We measure the memory footprint of SimCLR, versus MoCo-v2 (keeping all hyperparameters the same, except for the use of the momentum encoder). We compare ResNet-50 models with batch size 512, split over two V100s. We find that MoCo-v2 uses 16.3G of memory per GPU, compared to 15.4G for SimCLR, a 6% increase. The same scale of difference will be observed with CAN with a momentum encoder.
>
> We hope this information provides reassurance. Although we also considered running comparisons to alternative hybrid methods, we decided against it due to differences in setup (and in some cases lack of public code). Instead we opted for the cleanest comparison possible, to SimCLR and MAE. We do not believe that this choice loses too much, since other hybrid methods are at least as costly as SimCLR.

---

### Review · Reviewer_nvU1 · 2023-08-06

**Summary Of Contributions:**

This paper presents CAN, a hybrid self-supervised learning method that combines contrastive learning, masked autoencoders, and noise prediction from diffusion models. The paper demonstrates the proposed approach improves efficiency and performance, even surpassing popular methods like MAE and SimCLR.

**Audience:**

No

**Claims And Evidence:**

No

**Requested Changes:**

Kindly conduct a comprehensive review of the paper and ensure its completion.


**Strengths And Weaknesses:**

Weaknesses:

1. The paper appears to be incomplete, as key elements such as Figure 2 and Figure 3 are missing from the main text, and it is difficult to follow with Table ??, Figure ?? and Appendix ??. Navigating the material becomes challenging due to these gaps.
2. The innovation of the proposed technique seems limited, primarily combine three existing loss components rather than introducing a substantially novel approach.

---

> ### Author Response · Authors · 2023-08-20
> **Formatting is amended**
>
> Thank you for volunteering to review our work.
>
> We think that the current state of this review does not fairly represent our work, and would like to take some time to address your initial concerns. We hope that once these initial concerns are straightened out you will revisit our work and provide more feedback on the content of the paper.
>
> ---
>
> > The paper appears to be incomplete
>
> We do not agree with this. You are, however, correct in noticing the existence of some broken references such as “Figure ??”. This is our mistake, and we apologise. We have uploaded a fixed version. The mistake occurred during the final upload of the paper. Only the broken links all refer to the appendix (and vice versa the appendix links that refer to the main paper) were broken. The paper itself is almost entirely self contained, with the only references to the appendix being to: 1) point readers to hyperparameter details and 2) to refer to an additional set of results on ImageNet-21k. We note that in both of these cases the correct information can still easily be found by matching words.
>
> We acknowledge this as a genuine mistake, and apologise for the inconvenience caused. However, claiming that the paper is _incomplete_ is completely out of proportion, and does not reflect the manuscript fairly. The fact that the other reviewers were able to make good-faith attempts to review our work speaks to this.
>
> Now we have an updated version of the manuscript, we ask you to revisit it and give our work a fair chance.
>
> ---
>
> > The innovation of the proposed technique seems limited, primarily combining three existing loss components rather than introducing a substantially novel approach.
>
>
> Whilst it is true that we combine three losses, this statement is reductive and does not reflect the central contribution of our work. We note that there are already a number of published works dedicated to combined self-supervised objectives. See for instance [Huang et al. 2022, Tao et al., 2022; Chen et al., 2022; Assran et al., 2022], all of which are cited in Section 2 (related work).
>
> Instead of simply proposing combining several losses, the core contribution of our work is to demonstrate that hybrid (contrastive masked-autoencoder) methods can be used to find favourable efficiency-performance trade-offs. That is, we introduce a method that is much more efficient that contrastive learning, but outperforms both contrastive learning and MAE).
>
> This is different from prior hybrid methods, which focus on maximising performance, paying little or no attention to the efficiency-performance trade-offs. Our contribution, therefore, is to show that by rethinking the search of the design space of hybrid methods it is possible to identify approaches that are both  _efficient_ and _performant_.
>
> ---
>
> Thank you again for volunteering to review our work, and we look forward to seeing your updated review.

---

### Decision · Action_Editors · 2023-09-30

**Recommendation:** Reject

**Comment:**

The reviewers acknowledged that the paper tackles an important topic. They appreciated improved downstream performance when we combine SimCLR and MAE as the pretraining objective.

However, there were two major concerns: 1) The main technical idea of combining SimCLR and MAE has been explored extensively in the community. I am not using this as the reason for rejection though -- this criticism has been already acknowledged by the authors in the paper, and they emphasized that the main contribution of the paper is not to propose this as a new technique, but rather showing that it improves efficiency. 2) There is insufficient evidence to fully support the claimed efficiency claim. As noted by `htve`, Figure 1 demonstrates that the proposed approach is less efficient than MAEs and more efficient than SimCLR. But there is no other experiment investigating the efficiency of the approach.

Since the focus of this paper is on efficiency, extensive experimental validation is necessary. Showing how the combined SimCLR & MAE objective scales with various factors --such as image resolution, number of tokens, batch size, dataset size, different compute capacity budgets (both during pretraining and finetuning stages), various downstream tasks including image (classification, segmentation, generation, etc.) and video (classification, language grounding, tracking, etc.), and many others.-- could greatly enhance the quality of the submission.

Given these limitations, we recommend rejection at this time.

**Audience:**

The reviewers acknowledged that improving the efficiency self-supervised approaches is an important topic, which should be of interest to the TMLR's audience.

**Claims And Evidence:**

As noted by the reviewers (most clearly stated by `htve`), the claimed contribution of improved efficiency of the proposed approach is not supported by convincing and clear evidence.